# A Point Cloud Registration Method Based on Histogram and Vector Operations

Yanan Zhang [1,2], Dayong Qiao [1,2,3,*], Changfeng Xia [4] and Qing He [5]

1 Key Laboratory of Micro/Nano Systems for Aerospace, Ministry of Education, Northwestern Polytechnical University, Xi'an 710072, China
2 Shaanxi Province Key Laboratory of Micro and Nano Electro-Mechanical Systems, Northwestern Polytechnical University, Xi'an 710072, China
3 Ningbo Institute of Northwestern Polytechnical University, 218 Qingyi Road, Ningbo 315103, China
4 Xi'An Zhisensor Technologies Co., Ltd., Xi'an 710077, China
5 Konka Group Company Limted, Kejinan 12 Road 28, Shenzhen 518063, China
* Correspondence: dyqiao@nwpu.edu.cn

**Abstract:** Point-pair registration in a real scene remains a challenging task, due to the complexity of solving three transformations (scale, rotation, and displacement) simultaneously, and the influence of noise and outliers. Aimed at this problem, a registration algorithm based on histogram and vector operations is proposed in this paper. This approach converts point-based operations into vector-based operations, thereby decomposing the registration process into three independent steps solving for scale transformation factors, rotation matrices, and displacement vectors, which reduces the complexity of the solution and avoids the effects of scaling in the other two processes. The influence of outliers on the global transformation matrix is simultaneously eliminated using a histogram-based approach. Algorithm performance was evaluated through a comparison with the most commonly used SVD method in a series of validation experiments, with results showing that our methodology was superior to SVD in the cases with scaling transformation or outliers.

**Keywords:** point-pair registration; vector-based operation; histogram-based operation

## 1. Introduction

The use of 3D point cloud processing has seen rapid expansion over the last several years, promoted by developments in computer hardware and increased processing power. Recent applications have included a broad range of fields, such as industrial production, autonomous driving, and cultural relic restoration [1–3]. However, it remains difficult to completely reconstruct objects using a single depth sensor, due to field-of-view limitations and occlusions. As such, multiple depth sensors are typically required to measure an object from multiple angles and positions. Reconstructed results can then be converted to the same coordinate system and fused into a complete point cloud [4]. This process requires point cloud registration, a challenging task in pattern recognition [5], computer vision [6], robotics [7], and industrial measurements [8]. Its purpose is to identify a transformation matrix from one coordinate system to another, allowing the point clouds from different sensors to be converted into the same coordinate system, thereby forming a complete point cloud for the measured object [9–11].

Point cloud registration algorithms are typically implemented in three steps. The first step involves determining the correspondence of points between two sets. The second step requires calculating a transformation matrix between the two point sets using this correspondence. The third step applies iterative methods to optimize the transformation matrix [12]. The first step can be achieved using a variety of techniques, including distance-based [13] and feature-based methods [14]. The third step can primarily be divided into optimization methods based on distance [15], annealing, and soft correspondence [16],

similarity measurements [17], and probability density estimation [18]. In terms of the second step, most techniques use an algorithm based on SVD (singular value decomposition) to calculate the transformation matrix [19–23]. This approach is not only fast, but it also produces the smallest errors when only rotation and translation transformations are applied between point sets [24].

When only rotation and translation operations are conducted between point sets, the algorithm can achieve good results. However, It is very difficult for the SVD algorithm to identify and handle the scale transformation. Here, scale transformation is a linear transformation that enlarges or shrinks objects by a scale factor that is the same in all directions. In practice, the sizes of two reconstructed point clouds are always different. Therefore, if scale transformation is neglected, not only can the transformation relationship of two point clouds not be correctly expressed, but also unavoidable error will be introduced into rotation and translation processes [25]. These issues are the result of errors and variability in different sensors, which are essentially unavoidable. In addition, all points are involved in calculating the transformation matrix and, as such, calculation errors increase when local noise is too large or outliers are included [26]. Therefore, point clouds must be filtered before this registration process, and it has high requirements for filtering [27–29].

In response to the above problems, scholars have proposed various solutions. The PFH (point feature histogram) algorithm proposed by Radu computes the registration results by extracting an optimal set that best characterizes a given point cloud through feature histograms [30]. This method is insensitive to outliers, but the computational efficiency is generally low due to the high computational dimension (16D). Baowei utilizes the PCA (Principal Component Analysis) algorithm to calculate the covariance matrix of the source and target point clouds, and finds the rotation and transformation matrices between the source and target point clouds based on the covariance matrix [25]. This method works well in the case of scale transformation, but the computation time is also long, taking several minutes to compute 100,000 points. In addition, Xuyan's proposed method based on triangle similarity ratio consistency uses LASH (Local Angle Statistics Histogram) to detect triangle matching points with the same similarity for computing multiple transformations between two point clouds, which has been proved to be robust to noise [31]. Xu's method applies the NVP (normal vector and particle swarm optimization) algorithm to find the correspondence points and employs the quaternion method to align the correspondence points, which achieves high accuracy even when some data are lost [32]. Yet, this method generally requires a long time when evaluating multiple transformation matrices.

Aimed at this complex registration process, this study proposes a registration method based on histogram and vector operations. To avoid the mutual influence of registration parameters, our method converts point-based operations into vector-based operations, thereby decomposing the registration process into three independent steps solving for scale transformation factors, rotation matrices, and displacement vectors, which reduces the complexity of the solution. In case of noise and outlier situations, histograms are utilized to obtain Rodrigues parameters (i.e., scale factor, rotation axis vector, sine and cosine values of the rotation angle), following the processes to calculate the transformation matrices based on Rodrigues' rotation formula. Finally, four experiments demonstrate that the proposed method has good registration accuracy both in the case of scale transformation and outliers.

Compared to existing methods, the novelty of this paper is: (i) Other methods address the scale and outlier problem in terms of feature extraction, and our method addresses both of these problems in the computation of the transformation matrix. (ii) We calculated the scale factor, rotation axis vector, rotation angle, and displacement vector separately using vector and histogram methods to reduce the error interactions of these variables. (iii) We make full use of the relationship between vectors (Equations (5)–(8)) to calculate the rotation axis vector and the rotation angle. (iv) Even under Gaussian noise, the distribution of Rodrigues parameters will not be the standard positive-terrestrial distribution, which makes the error of methods such as averaging very large. In contrast, our method uses

histograms to greatly avoid this systematic error and provides a way for others to find the maximum likelihood variable for non-Gaussian distributions.

This paper is organized as follows. The principles of the proposed registration methodology are introduced in Section 2. Experimental details are provided in Section 3. The discussions are presented in Section 4. The conclusion is given in the last section.

## 2. Methods

### 2.1. Symbol Description

Since many symbols are used in this paper, Table 1 explains the meaning of these symbols.

**Table 1.** The list of symbols used in this paper.

| Symbol | Description |
|---|---|
| $\{p_i\}$ | The initial point cloud |
| $\{p_i'\}$ | The target point cloud |
| $p_i$ | a point in the point cloud $\{p_i\}$ |
| $p_i'$ | The corresponding point of $p_i$ in point cloud $\{p_i'\}$. |
| $\vec{p}_i$ | The coordinates of point $p_i$ |
| $\vec{p}_i'$ | The coordinates of point $p_i'$ |
| $H$ | Homogeneous transformation matrix between $\{p_i\}$ and $\{p_i'\}$. |
| $I$ | $3 \times 3$ unit matrix. |
| $\vec{n}$ | The rotation axis of $\{p_i\}$ transformed to $\{p_i'\}$ |
| $(k_x, k_y, k_z)$ | The coordinates of the rotation axis $\vec{n}$ |
| $K$ | The anti-symmetric matrix representation of $(k_x, k_y, k_z)$. |
| $\theta$ | The rotation angle of $\{p_i\}$ transformed to $\{p_i'\}$ |
| $R$ | The rotation matrix of $\{p_i\}$ transformed to $\{p_i'\}$ |
| $R_{real}$ | The real value of $R$ |
| $R_{calculate}$ | The calculated value of $R$ |
| $\vec{t}$ | The translation vector of $\{p_i\}$ transformed to $\{p_i'\}$ |
| $\vec{t}_{real}$ | The real value of $\vec{t}$ |
| $\vec{t}_{calculate}$ | The calculated value of $\vec{t}$ |
| $scale$ | The scale factor of $\{p_i\}$ transformed to $\{p_i'\}$ |
| $scale_{real}$ | The real value of $scale$ |
| $scale_{calculate}$ | The calculated value of $scale$ |
| $A\ B\ C$ | Three points in the point cloud $\{p_i\}$ |
| $A'\ B'\ C'$ | The corresponding points of points $A$, $B$ and $C$ in $\{p_i'\}$. |
| $\overrightarrow{axis_{ABC}}$ | The rotation axis of $A\ B\ C$ transformed to $A'\ B'\ C'$ |
| $\overrightarrow{p_m p_n}$ | The vector from point $p_m$ to point $p_n$ |
| $\|\overrightarrow{p_m p_n}\|$ | The norm of vector $\overrightarrow{p_m p_n}$ |
| $\times$ | Cross product of vectors |
| $\bullet$ | Dot product of vectors |
| $normalize()$ | Normalization of vectors |
| $\varepsilon$ | The accuracy of registration algorithms |
| $\sigma$ | Gaussian noise standard deviation |

### 2.2. Understanding of the Registration Process

We all know that the registration process between two point clouds $\{p_i\}$ and $\{p_i'\}$ is actually to find the transformation matrix $H$, which makes the corresponding points $p_i$ and $p_i'$ between the two point clouds meet the following relationship:

$$p_i' = Hp_i \tag{1}$$

In homogeneous coordinates,

$$H = \begin{bmatrix} R & \vec{t} \\ \vec{0} & \frac{1}{scale} \end{bmatrix} \tag{2}$$

Among them, $R$, $\vec{t}$, and *scale* are the rotation matrix, displacement vector, and scale factor for transforming point cloud $\{p_i\}$ to point cloud $\{p_i'\}$. Therefore, the registration process of two point clouds is actually the process of solving the $R$, $\vec{t}$, and *scale* components of the two point cloud transformations.

### 2.3. Rodrigues' Rotation Formula

Rodrigues' rotation formula states that if a point cloud $\{p_i\}$ is rotated by an angle $\theta$ around the unit vector $\vec{n}$ to form the point cloud $\{p_i'\}$, the associated rotation matrix $R$ can be represented by:

$$R = I + (1 - \cos\theta)K^2 + \sin\theta K \tag{3}$$

Here, $I$ is a $3 * 3$ unit matrix and $K$ can be expressed as:

$$K = \begin{bmatrix} 0 & -k_z & k_y \\ k_z & 0 & -k_x \\ -k_y & k_x & 0 \end{bmatrix} \tag{4}$$

where the coordinates of the rotation axis $\vec{n}$ are denoted $(k_x, k_y, k_z)$. Once $\vec{n}$ and the rotation angle $\theta$ are determined, the rotation matrix $R$ can be calculated from the point cloud $\{p_i\}$ to the point cloud $\{p_i'\}$.

### 2.4. Registration of Three Pairs of Corresponding Points

Three points ($A$, $B$, and $C$) can be identified in a given point cloud $\{p_i\}$, corresponding to three points ($A'$, $B'$, and $C'$) in the point cloud $\{p_i'\}$. A scale transformation factor from points $A$, $B$, and $C$ to points $A'$, $B'$, and $C'$ can then be calculated as:

$$scale = \frac{\|\overrightarrow{A'B'}\|}{\|\overrightarrow{AB}\|} = \sqrt{\frac{(x_A' - x_B')^2 + (y_A' - y_B')^2 + (z_A' - z_B')^2}{(x_A - x_B)^2 + (y_A - y_B)^2 + (z_A - z_B)^2}} \tag{5}$$

where $(x_A, y_A, z_A)$, $(x_B, y_B, z_B)$, $(x_A', y_A', z_A')$, and $(x_B', y_B', z_B')$ are the coordinates of points $A$, $B$, $A'$ and $B'$, respectively. The axis of the rotation vector can be expressed as:

$$\overrightarrow{axis_{ABC}} = normalize\left(\left(\frac{\overrightarrow{AB}}{\|\overrightarrow{AB}\|} - \frac{\overrightarrow{A'B'}}{\|\overrightarrow{A'B'}\|}\right) \times \left(\frac{\overrightarrow{AC}}{\|\overrightarrow{AC}\|} - \frac{\overrightarrow{A'C'}}{\|\overrightarrow{A'C'}\|}\right)\right) \tag{6}$$

Since the unit vector $\frac{\overrightarrow{AB}}{\|\overrightarrow{AB}\|}$ is rotated around the rotation axis $\overrightarrow{axis_{ABC}}$ to the vector $\frac{\overrightarrow{A'B'}}{\|\overrightarrow{A'B'}\|}$. The vector $\overrightarrow{axis_{ABC}}$ is then perpendicular to the vector $\frac{\overrightarrow{AB}}{\|\overrightarrow{AB}\|} - \frac{\overrightarrow{A'B'}}{\|\overrightarrow{A'B'}\|}$ and $\overrightarrow{axis_{ABC}}$ is also perpendicular to $\frac{\overrightarrow{AC}}{\|\overrightarrow{AC}\|} - \frac{\overrightarrow{A'C'}}{\|\overrightarrow{A'C'}\|}$. Thus, Equation (6) can be used to calculate the unit vector $\overrightarrow{axis_{ABC}}$, where $normalize()$ represents a normalization operation. The cosine of the angle $\theta$ is given by:

$$\cos\theta = \frac{\left(\frac{\overrightarrow{AB}}{\|\overrightarrow{AB}\|} - \frac{\overrightarrow{A'B'}}{\|\overrightarrow{A'B'}\|} \bullet \overrightarrow{axis_{ABC}}\right) \bullet \left(\frac{\overrightarrow{AC}}{\|\overrightarrow{AC}\|} - \frac{\overrightarrow{A'C'}}{\|\overrightarrow{A'C'}\|} \bullet \overrightarrow{axis_{ABC}}\right)}{\|\frac{\overrightarrow{AB}}{\|\overrightarrow{AB}\|} - \frac{\overrightarrow{A'B'}}{\|\overrightarrow{A'B'}\|} \bullet \overrightarrow{axis_{ABC}}\| \cdot \|\frac{\overrightarrow{AC}}{\|\overrightarrow{AC}\|} - \frac{\overrightarrow{A'C'}}{\|\overrightarrow{A'C'}\|} \bullet \overrightarrow{axis_{ABC}}\|} \tag{7}$$

The sine value of the angle $\theta$ is given by:

$$\sin\theta = \frac{\left(\frac{\overrightarrow{AB}}{\|\overrightarrow{AB}\|} - \frac{\overrightarrow{A'B'}}{\|\overrightarrow{A'B'}\|} \bullet \overrightarrow{axis_{ABC}}\right) \times \left(\frac{\overrightarrow{AC}}{\|\overrightarrow{AC}\|} - \frac{\overrightarrow{A'C'}}{\|\overrightarrow{A'C'}\|} \bullet \overrightarrow{axis_{ABC}}\right)}{\|\frac{\overrightarrow{AB}}{\|\overrightarrow{AB}\|} - \frac{\overrightarrow{A'B'}}{\|\overrightarrow{A'B'}\|} \bullet \overrightarrow{axis_{ABC}}\| \cdot \|\frac{\overrightarrow{AC}}{\|\overrightarrow{AC}\|} - \frac{\overrightarrow{A'C'}}{\|\overrightarrow{A'C'}\|} \bullet \overrightarrow{axis_{ABC}}\|} \bullet \overrightarrow{axis_{ABC}} \tag{8}$$

When calculating the rotation angle of two vectors, the components in the direction of the rotation axis must be removed first. The angle can then be acquired using Equations (7) and (8) and the rotation matrix $R$ can be calculated from Equation (3). The displacement vector $\vec{t}$ from points $A$, $B$, and $C$ to points $A'$, $B'$, and $C'$ can be calculated as:

$$\vec{t} = \frac{1}{3} \sum_{i=1}^{3} (\frac{1}{scale} \cdot \vec{p_i'} - R \cdot \vec{p_i}) \tag{9}$$

where $\vec{p_i}(i = 1, 2, 3)$ represents the coordinates of points $A$, $B$, and $C$, respectively, and $\vec{p_i'}(i = 1, 2, 3)$ denotes the coordinates of points $A'$, $B'$, and $C'$, respectively.

### 2.5. Registration of Two Sets of Point Clouds

Point clouds can be grouped using three points, which can uniquely determine a registration relationship. The global registration relationship can then be determined from the registration relationship of each group as follows.

- Step 1: The point clouds $\{p_i\}$ and $\{p_i'\}$ are separately grouped using three points.
- Step 2: The technique presented in Section 2.3 is used to calculate the Rodrigues parameters (i.e., scale factor, rotation axis vector, and sine and cosine values of the rotation angle) for each group.
- Step 3: A histogram is used to acquire the global Rodrigues parameters, taking the parameter with the highest probability in the histogram as the global parameter.
- Step 4: A global rotation matrix is acquired using the global Rodrigues parameters.
- Step 5: A global displacement vector is calculated using:

$$\vec{t} = \frac{1}{N} \sum_{i=1}^{N} (\frac{1}{scale} \cdot \vec{p_i'} - R \cdot \vec{p_i}) \tag{10}$$

where $\vec{p_i}$ represents the coordinates of the point $p_i$ in the point cloud $\{p_i\}$ and $\vec{p_i'}$ represents the coordinates of the point $p_i'$ in the point cloud $\{p_i'\}$.

## 3. Experiments

Algorithm effectiveness was evaluated on the i5-10400 platform. We used a point cloud from the Bremen dataset as the source point cloud $\{p_i\}$, which can be seen in Figure 1. Then, we transform the original point cloud according to Equation (11) to obtain the destination point cloud $\{p_i'\}$. Registration processing was performed on the two sets of point clouds using our proposed technique and the SVD algorithm.

$$\vec{p_i'} = scale_{real} \cdot (R_{real} \cdot \vec{p_i} + \overrightarrow{t_{real}}) \tag{11}$$

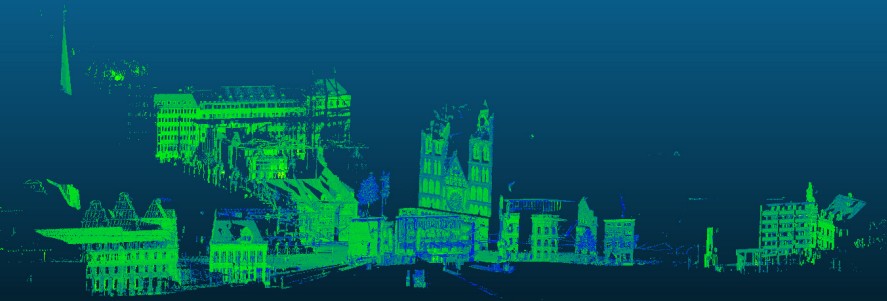

**Figure 1.** The Bremen point cloud used in the validation experiments.

The accuracy of the two algorithms was determined by calculating the error using the following formula:

$$\varepsilon = \frac{1}{N} \sum_{i=1}^{N} (\frac{1}{scale_{calculate}} \cdot \vec{p'_i} - R_{calculate} \cdot \vec{p_i} - \overrightarrow{t_{calculate}})^2 \tag{12}$$

where $scale_{calculate}$, $R_{calculate}$, and $\overrightarrow{t_{calculate}}$ are the scale factor, rotation matrix, and displacement vector calculated using our method and the SVD algorithm, respectively. The default SVD scaling factor was set to $scale_{calculate} = 1$. Algorithm performance was verified in four distinct cases:

Case 1: Ideal conditions (no scaling transformation, noise, or outliers).

Case 2: Noisy conditions (no scaling transformation or outliers, but noise included).

Case 3: Scaled conditions (no outliers, but noise and a scaling transformation included).

Case 4: Outlier conditions (no scaling transformation, but noise and outliers included).

In Equation (11), the rotation matrix and displacement vector of the original point cloud transformation to the target point cloud are:

$$R_{real} = \begin{bmatrix} 0.3536 & 0.3536 & 0.866 \\ -0.7071 & 0.7071 & 0 \\ -0.6124 & -0.6124 & 0.5 \end{bmatrix} \tag{13}$$

$$\overrightarrow{t_{real}} = \begin{bmatrix} 100 & 200 & 50 \end{bmatrix} \tag{14}$$

When there is no scale transformation, we define the factor of the scale transformation as 1. When there is scale transformation, we define the range of scale transformation as 0.9–1.1. In addition, the Gaussian noise standard deviation in this paper is 0.2, and the outliers are square point clouds with side lengths of $14 * 14$ and spacing of 0.1.

### 3.1. The Case of No Scale Transformation and No Noise

In this case, the transformations from the origin point cloud to the destination point cloud are rotation and translation transformations. The corresponding transformed point cloud $\{p'_i\}$ is shown in Figure 2. Then, we calculated the registration parameters according to the steps in Section 2.5.

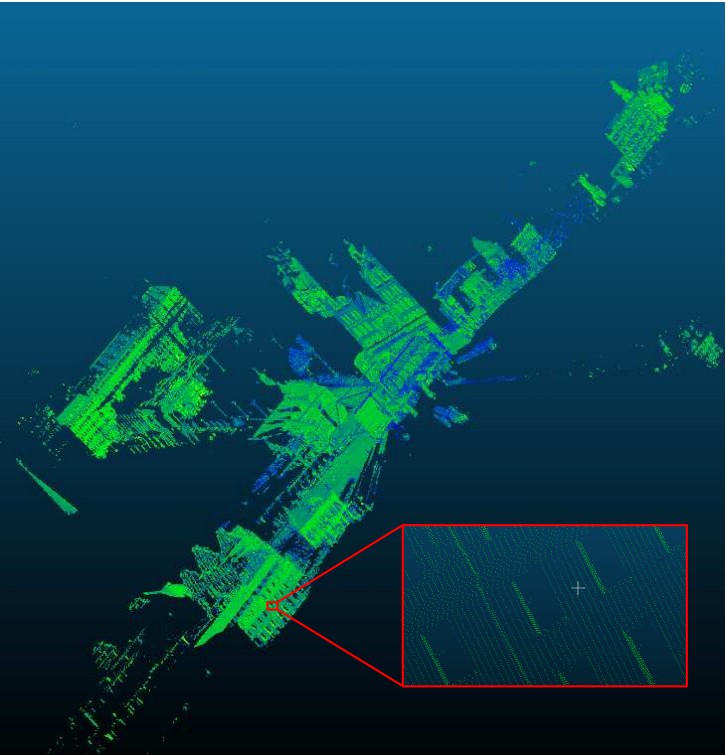

**Figure 2.** The point cloud transformed by rotation and translation.

After the grouping process in step 1 and the vector operation in step 2, histograms for the scaling factor, cosine, and sine values of the rotation angle and rotation axis vector in step 3 are shown in Figure 3. We took the value of the maximum probability from the histogram of each Rodrigues parameter in step 3 as the global Rodrigues parameter and calculated the rotation matrix of the registration by Equation (3) (step 4) as:

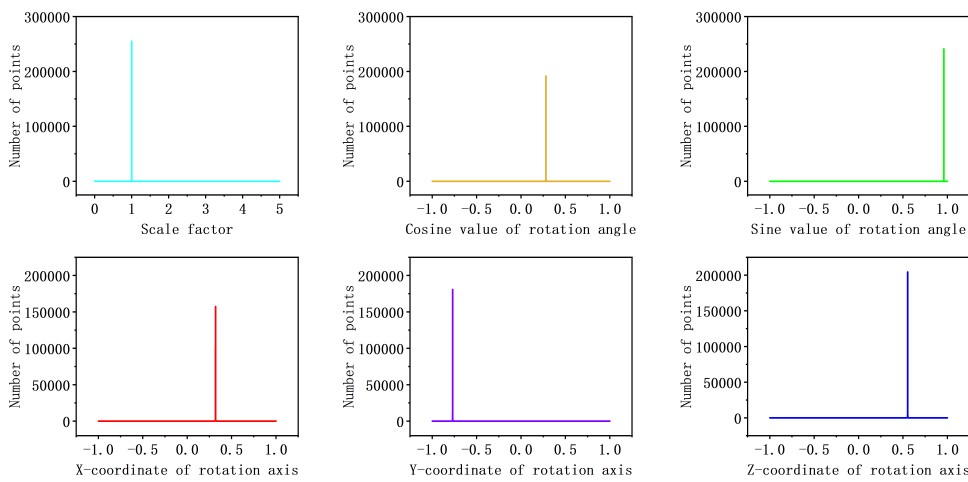

**Figure 3.** Histograms for the Rodrigues parameters calculated using our method in Case 1.

$$R_{calculate} = \begin{bmatrix} 0.3536 & 0.3535 & 0.866 \\ -0.707 & 0.7072 & 0 \\ -0.6124 & -0.6123 & 0.5 \end{bmatrix} \tag{15}$$

In step 5, we calculated the registered displacement vector by Equation (10) as:

$$\overrightarrow{t_{calculate}} = \begin{bmatrix} 100 & 200 & 50 \end{bmatrix} \tag{16}$$

and those calculated using SVD are given by:

$$R_{calculate} = \begin{bmatrix} 0.3536 & 0.3536 & 0.866 \\ -0.7071 & 0.7071 & 0 \\ -0.6124 & -0.6124 & 0.5 \end{bmatrix} \tag{17}$$

$$\overrightarrow{t_{calculate}} = \begin{bmatrix} 100 & 200 & 50 \end{bmatrix} \tag{18}$$

### 3.2. The Case without a Scaling Transformation but with Gaussian Noise

Gaussian noise with a standard deviation of two times the point spacing ($\sigma = 0.2$) was added to the destination point cloud $\{p_i'\}$ described in Section 3.1. The destination point cloud $\{p_i'\}$ with Gaussian noise is shown in Figure 4. Then, we calculated the registration parameters according to the steps in Section 2.5. In this case, histograms for the scaling factor, cosine, and sine values of the rotation angle and rotation axis vector are shown in Figure 5.

The scaling factor was set to 1 and the rotation matrix and displacement vector calculated using our method are:

$$R_{calculate} = \begin{bmatrix} 0.3546 & 0.3539 & 0.8655 \\ -0.7066 & 0.7077 & 0 \\ -0.6124 & -0.6115 & 0.501 \end{bmatrix} \tag{19}$$

$$\overrightarrow{t_{calculate}} = \begin{bmatrix} 100.0063 & 199.9923 & 49.9844 \end{bmatrix} \tag{20}$$

Those produced by the SVD method are:

$$R_{calculate} = \begin{bmatrix} 0.3536 & 0.3536 & 0.866 \\ -0.7071 & 0.7071 & 0 \\ -0.6124 & -0.6124 & 0.5 \end{bmatrix} \tag{21}$$

$$\overrightarrow{t_{calculate}} = \begin{bmatrix} 100.0005 & 200.0001 & 49.9999 \end{bmatrix} \tag{22}$$

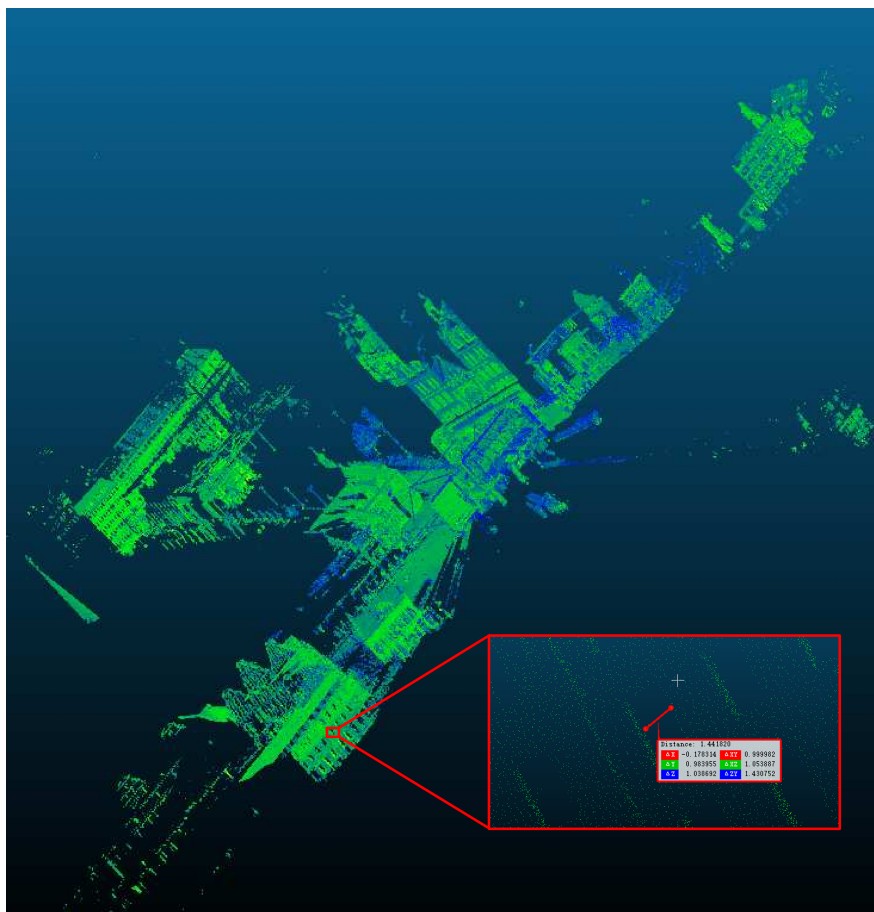

**Figure 4.** The destination point cloud $\{p_i'\}$ with noise.

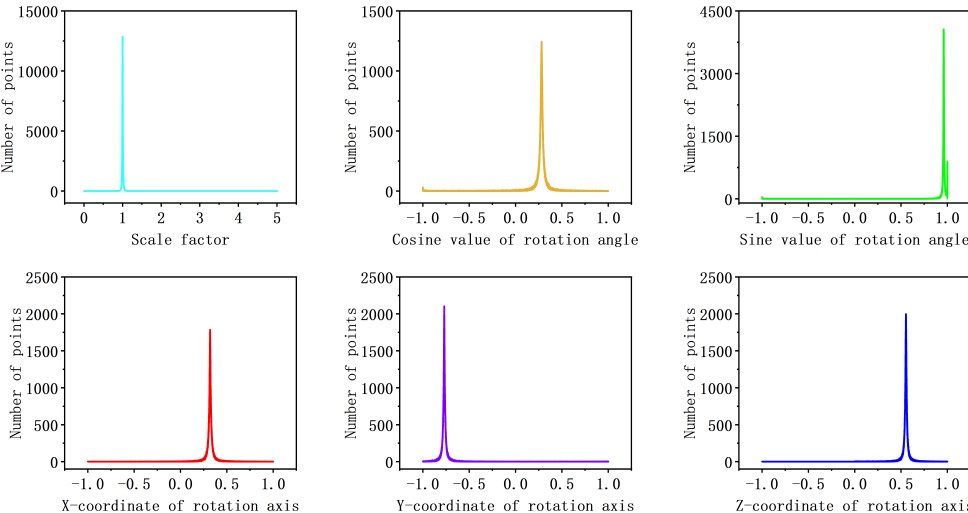

**Figure 5.** Histograms for the Rodrigues parameters calculated using our method in Case 2.

### 3.3. The Case with Scaling Information and Gaussian Noise

The scaling effects produced by each of these two methods were observed using scale factors ($scale_{real}$) ranging from 0.9 to 1.1, applied to the point cloud $\{p'_i\}$ discussed in Section 3.2. The destination point cloud $\{p'_i\}$ with Gaussian noise and scale transformation is shown in Figure 6. We then registered the point clouds $\{p_i\}$ and $\{p'_i\}$ using each technique. Among them, when the scale factor is 0.9, histograms for the scaling factor, cosine, and sine values of the rotation angle and rotation axis vector are shown in Figure 7.

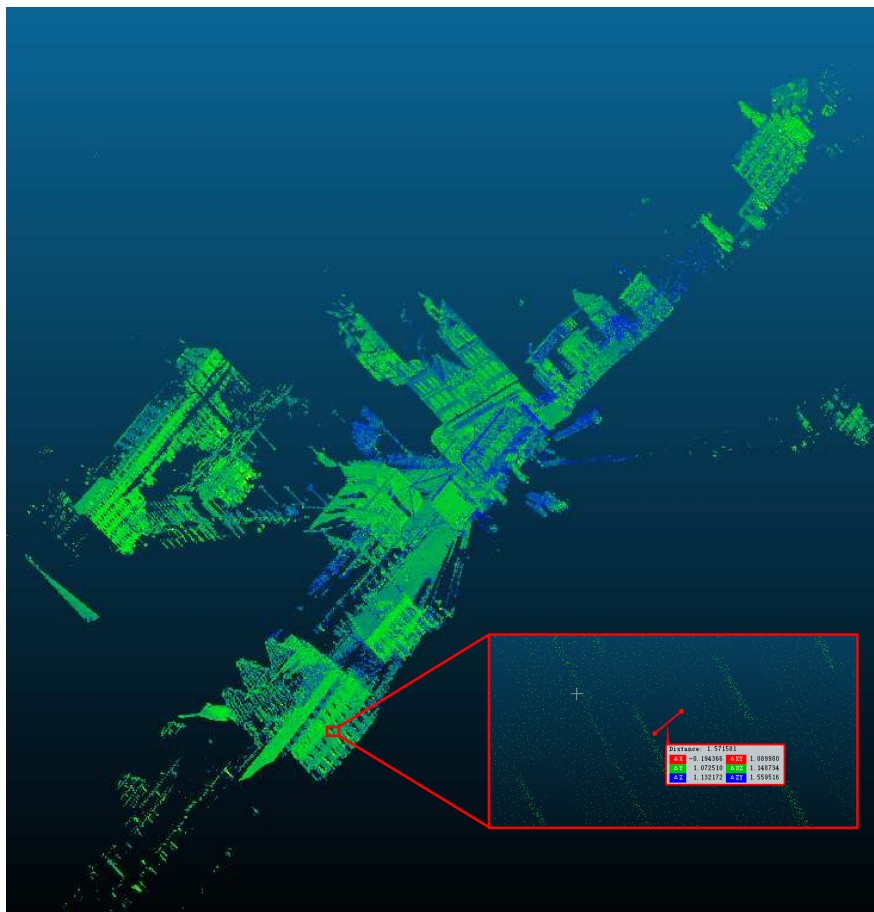

**Figure 6.** The destination point cloud $\{p'_i\}$ with scale and noise.

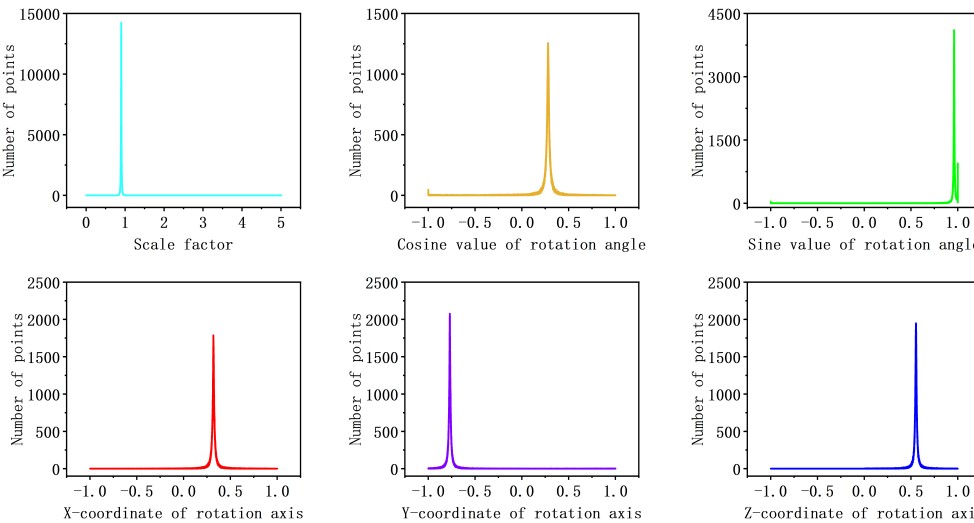

**Figure 7.** Histograms for the Rodrigues parameters calculated using our method in Case 3.

The scale factor calculated in step 3 is:

$$scale_{calculate} = 0.9 \tag{23}$$

The rotation matrix calculated in step 4 is:

$$R_{calculate} = \begin{bmatrix} 0.3538 & 0.3531 & 0.8659 \\ -0.7067 & 0.7074 & 0 \\ -0.6124 & -0.6122 & 0.4999 \end{bmatrix} \tag{24}$$

The displacement vector calculated in step 5 is:

$$\overrightarrow{t_{calculate}} = \begin{bmatrix} 100.0035 & 199.9940 & 50.0002 \end{bmatrix} \tag{25}$$

The rotation matrix and displacement vector produced by the SVD method are:

$$R_{calculate} = \begin{bmatrix} 0.3484 & 0.3578 & 0.8664 \\ -0.7095 & 0.7047 & -0.0057 \\ -0.6125 & -0.6127 & 0.4994 \end{bmatrix} \tag{26}$$

$$\overrightarrow{t_{calculate}} = \begin{bmatrix} 90.7114 & 180.9782 & 45.3664 \end{bmatrix} \tag{27}$$

The results for varying scale factors are shown in Figure 8.

It can be seen in Figure 8a that in the process of increasing the scaling factor from 0.9 to 1.1, the maximum error produced by our method is 0.05% and close to zero in most cases. Errors calculated using both our method and SVD are shown in Figure 8b, where it is evident that the SVD method produced the smallest errors for a scaling factor equal to 1, increasing as the scale factor deviated from 1 (e.g., reaching 60 for a scale factor of 1.1). As shown, our method is far less sensitive to changes in the scaling factor.

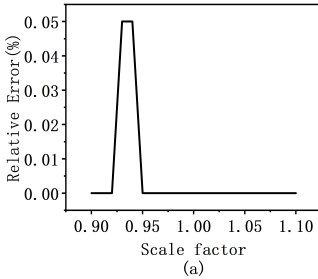
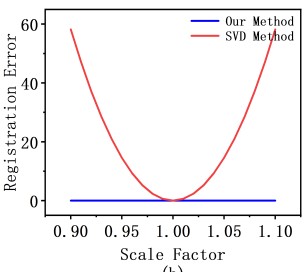

**Figure 8.** The experimental results for Case 3. (**a**) The relative error for varying scale factors, calculated using our method. (**b**) A comparison of errors produced using our method and SVD, applied to Case 3.

### 3.4. The Case with Outliers and Gaussian Noise

Some outliers will inevitably be generated in point clouds, due to the influence of noise, varying fields of view, and moving objects in the environment. To evaluate the impact on each technique, we added outliers (square point clouds with side lengths of $14 * 14$ and spacing of 0.1) to $\{p_i'\}$, as shown in Figure 9.

We then registered the source and destination point clouds using our method and SVD. Among them, the histogram in step 3 for the proposed technique is shown in Figure 10.

The rotation matrix and displacement vector calculated in step 4 and step 5 can be represented as:

$$R_{calculate} = \begin{bmatrix} 0.3532 & 0.3544 & 0.866 \\ -0.7079 & 0.7064 & 0 \\ -0.6119 & -0.6128 & 0.5003 \end{bmatrix} \tag{28}$$

$$\overrightarrow{t_{calculate}} = \begin{bmatrix} 99.9914 & 200.0110 & 49.9973 \end{bmatrix} \tag{29}$$

Those produced by the SVD method are given by:

$$R_{calculate} = \begin{bmatrix} 0.3484 & 0.3578 & 0.8663 \\ -0.7095 & 0.7046 & -0.0057 \\ -0.6125 & -0.6127 & 0.4994 \end{bmatrix} \tag{30}$$

$$\overrightarrow{t_{calculate}} = \begin{bmatrix} 99.9515 & 200.0389 & 50.0006 \end{bmatrix} \tag{31}$$

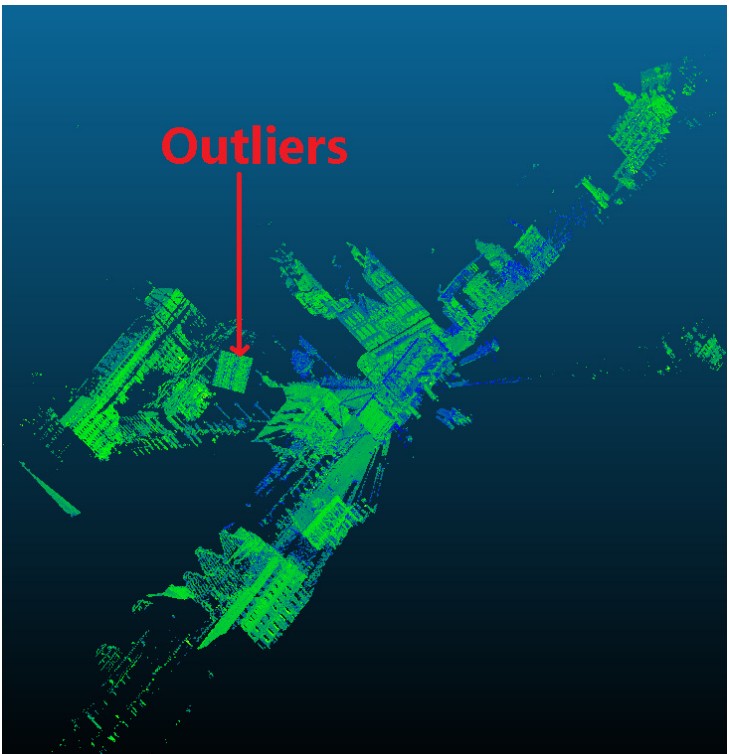

**Figure 9.** The point cloud $\{p_i'\}$ with outliers.

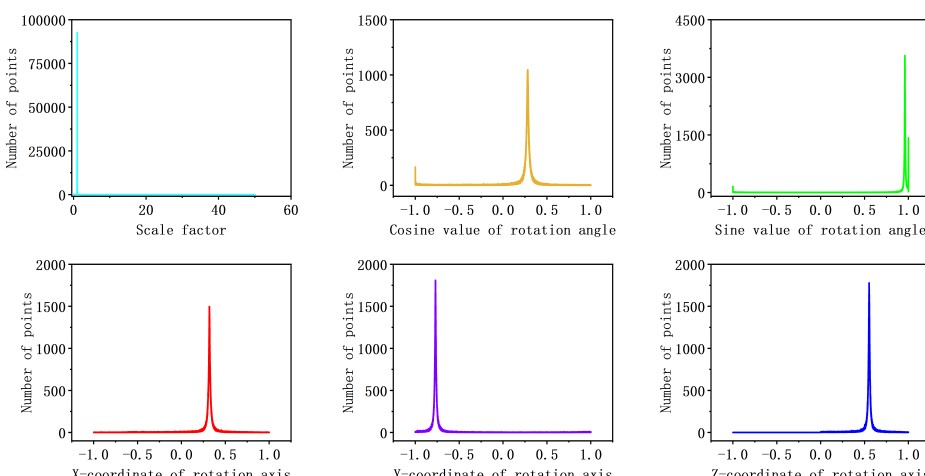

**Figure 10.** Histograms for the Rodrigues parameters calculated using our method in Case 4.

## 4. Results and Discussions

The experiments implemented above demonstrate that both our method and the SVD method are able to perform point cloud registration in each of the four cases. The results are shown in Table 2.

**Table 2.** Comparisons of our method and the SVD method.

| Experiment No. | 1 | 2 | 3 | 4 |
|---|---|---|---|---|
| Noise | No | Gaussian | Gaussian | Gaussian |
| Scaling | 1.0 | 1.0 | 0.9 | 1.0 |
| Outliers | No | No | No | Yes |
| Error of our method | $6.59 \times 10^{-5}$ | $2.70 \times 10^{-3}$ | 0.0017 | 0.0062 |
| Error of SVD method | 0 | $7.69 \times 10^{-7}$ | 58.1891 | 0.2541 |
| Time consumed by our method | 0.161 s | 0.178 s | 0.177 s | 0.159 s |
| Time consumed by SVD method | 0.011 s | 0.013 s | 0.015 s | 0.012 s |

As shown in Table 2, the SVD results are superior in ideal and noisy conditions and the errors of both methods are much smaller than the point spacing of $\{p_i\}$ and $\{p_i'\}$. This is because SVD is optimal for least squares errors, while our method suffers from histogram quantization effects. In the case of scale transformations, our method is better than the SVD method, because it cannot calculate scale transformation factors, which will eventually influence the registration results for angle and displacement transformations. It can be seen from Table 2 that even small-scale transformations could lead to large errors. In contrast, our method was unaffected by scale transformations.

The SVD method calculates a global (optimal) point cloud solution. As such, when mismatches (such as outliers) are included in the correspondence of the two sets, these point pairs participate in the calculation of point clouds. This can negatively affect the final registration results, as was the case for the fourth experiment. However, our method uses a statistical histogram approach involving a rotation axis vector, rotation angle, and scale transformation factor to determine the maximum probability for the global rotation axis, rotation angle, and scale transformation factor, thereby eliminating local outliers and noise. Compared with the SVD algorithm, which uses global points in the calculation of a transformation matrix, the influence of outliers on the registration process is reduced significantly, which improves the resulting accuracy. However, our technique does involve multiple vector operations and a time-consuming normalization step. As such, reducing the computational complexity and runtime will be the subject of a future study.

In addition, it is difficult to calculate scale, angle, and displacement transformations between point clouds while registering pairs of points with similar relationships. Thus, we convert these relationships into vectors, eliminating displacement transformations, and then normalize the vectors to eliminate scale transformations between the vectors. In this way, the purpose of separately calculating each transformation between point clouds is achieved, thereby reducing the difficulty. This study also utilizes the relationships between vectors to directly identify a rotation axis and rotation angle from a source point cloud to a destination point cloud, using Rodrigues' rotation formula to calculate the angle transformation matrix. This approach does not require a matrix decomposition or inversion, eliminating errors caused by failures in these steps.

## 5. Conclusions

Since SVD algorithm failed to solve the point-pair registration problem when scale transformation and outliers exist, a registration method based on histogram and vector operations is proposed in this paper. This method can reduce the complexity of point-pair registration by converting point-based operations into vector-based operations, and decomposing the registration process into three independent steps solving for scale transformation factors, rotation matrices, and displacement. This method can also eliminate the influence of outliers on the global transformation matrix by using a histogram-based approach. A series of validation experiments were carried out to validate the point-pair registration effect compared to the SVD-based method. The experimental results proved that our method has better performance in terms of relative error and registration error when scale transformations and outliers exist. Although our method has advantages in dealing with scale transformation and outliers, our method also has the disadvantage of

being time-consuming due to the involvement of vector operations. How to reduce the computational complexity of vector operations will be the focus of our future study. I believe that our method can become a reliable and efficient registration method in the field of 3D reconstruction and SLAM (Simultaneous Localization and Mapping) in the future.

**Author Contributions:** Methodology, Y.Z.; experiments, Y.Z.; writing—original draft preparation, Y.Z.; writing—review and editing, D.Q., C.X. and Q.H.; supervision, D.Q. All authors have read and agreed to the published version of the manuscript.

**Funding:** This research received support from the National Natural Science Foundation of China (Grant No. U21B2035, 62074128) and the Key-Area Research and Development Program of Guangdong Province (Grant No. 2021B0101410001).

**Institutional Review Board Statement:** Not applicable.

**Informed Consent Statement:** Not applicable.

**Data Availability Statement:** Not applicable.

**Conflicts of Interest:** The authors declare no conflict of interest.

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
