# Peer review of "A Point Cloud Registration Method Based on Histogram and Vector Operations"

_electronics, doi:10.3390/electronics11244172_

Round 1

Reviewer 1 Report

The authors have presented an A Point Cloud Registration Method Based on Histogram and Vector Operations, which involves too much mathematics. Based on the readings, I have some doubts.

1. Too much of mathematics is involved in the manuscript. The authors are advised to add a detailed symbol table that clears the readers' viewpoint.

2. What is the significance of scaling in Eq. 3. Add it into the revised manuscript.

3. Authors have not mentioned the proposed scheme's experimental setup (platform, model, simulation parameters). 

4. It is also advised to add a detailed dataset description in the experiments section.

5. The description of Figure 1 is missing in the manuscript. Add it to the revised manuscript.

6. Eqs 11-16 are not explained in the paper properly. Explain it properly for better understanding and readability.

7. It is advised to add a future scope of the work to the revised manuscript.

Author Response

We gratefully thank the editor and all reviewers for their time spent making their constructive remarks and useful suggestions, which has significantly raised the quality of the manuscript and enabled us to improve the manuscript. Each suggested revision and comment, brought by the reviewers was accurately incorporated and considered. Specific revisions and responses to each comment are provided in detail below.

Q1. Too much of mathematics is involved in the manuscript. The authors are advised to add a detailed symbol table that clears the readers' viewpoint.

Thank you very much for your suggestion, Table 1 lists the detailed symbols used in this paper.

Q2. What is the significance of scaling in Eq. 3. Add it into the revised manuscript.

Thank you for your advice. The meaning and significance of scaling are explained in the third paragraph of the introduction section.

Scaling is a linear transformation that enlarges or shrinks objects by a scale factor that is the same in all directions. In practice, the sizes of two reconstructed point clouds are always different. Therefore, if scale transformation is neglected, not only can the transformation relationship of two point clouds not be correctly expressed, but also unavoidable error will be introduced into rotations and translations process.

Q3. Authors have not mentioned the proposed scheme's experimental setup (platform, model, simulation parameters). 

Thank you very much for the reminder. The effectiveness of our algorithm and SVD algorithm was evaluated on the i5-10400 platform (It has been added to the experimental section). The codes are based on our method and the SVD method(Sorkine’s method), respectively.    

Q4. It is also advised to add a detailed dataset description in the experiments section.

Thank you very much for your suggestion. We added descriptions to the dataset in the experimental section (before Section 3.1) to ensure that the experiment could be replicated. The original point cloud we used is the Bremen dataset (This is a common dataset used in SLAM modeling and can be downloaded from the web). The transformation relationship between the original point cloud and the target point cloud is: Rreal, treal,  scalereal. The Gaussian noise standard deviation is 0.2, and the outliers are square point clouds with side lengths of 14*14 and spacing of 0.1.

Q5. The description of Figure 1 is missing in the manuscript. Add it to the revised manuscript.

Thank you for your reminder. Figure 1 shows the original point cloud we used, which we have added to the description at the beginning of the experiment.

Q6. Eqs 11-16 are not explained in the paper properly. Explain it properly for better understanding and readability.

We are very sorry for the reading discomfort caused to you. We have added appropriate content to the paper to ensure that the readers can understand Eqs 11-16.

In Section 2.2, we explained in detail that the process of registering two point clouds is actually the process of solving the R, t  and scale components of the two point cloud transformations. In case1, scale=0. Therefore, we only need to focus on the registration accuracy of  R and t. Equations 11-12 (now Equations 13-14) are the real values of  R and t. Equations 13-16 (now Equations 15-18) are  R and  t calculated by our method and SVD method, respectively. In Section 3.1, we take case 1 as an example to illustrate the solution steps in Section 2.5 of these components in detail.

After the grouping process in step 1 and the vector operation in step 2, histograms for the scaling factor, cosine, and sine values of the rotation angle and rotation axis vector in step 3 are shown in Fig. 3. We took the value of the maximum probability from the histogram of each Rodrigues parameter in step 3 as the global Rodrigues parameter and calculated the rotation matrix(Equations 15) of the registration by equation 3 (step 4) . In step 5, we calculated the registered displacement vector (Equations 16) by Equation 10.

Q7. It is advised to add a future scope of the work to the revised manuscript.

We totally agree with your suggestion. We have added the future scope of our work to the end of the conclusion section.

Although our method has advantages in dealing with scale transformation and outliers, our method also has the disadvantage of time-consuming due to the involvement of vector operations. How to reduce the computational complexity of vector operations will be the focus of our future study. I believe that our method can become a reliable and efficient registration method in the field of 3D reconstruction and SLAM(Simultaneous Localization and Mapping) in the future.

Finally, we would like to thank the referee again for taking the time to review our manuscript.

Reviewer 2 Report

This work provides a method for registering point clouds based on a histogram approach and vector operations. The research topic has interesting applications in the field of computer vision, and the article's presentation is comprehensible and engaging.

Before the paper may be judged fit for publication, I have just the following comments for the authors to address.

1. The literature review section of this article is quite deficient. After the introduction, the authors must create a separate section to describe in depth the state of the relevant literature in relation to their research. For instance, the authors only addressed the alternative strategy based on the SVD method, but there are other more modern approaches, such as those based on feature histogram, PCA, triangle similarity ration consistency, as well as normal vector and particle swarm optimization. Consequently, why were these existing methods not discussed?

2. Due to the lack of relevant literature, it is difficult to comprehend the basis for the ideas presented in this article. Are the histogram and vector operations utilized in this study completely novel? I have my doubts, as there are already studies that apply these methods, so what is genuinely novel about this paper?

3. In the results section, please include the images of the two point clouds for each case study. The authors only provided images for Case 1 and partially for Case 2. Please provide images for all case studies associated with each point cloud. By giving these images, readers will be able to comprehend the amount of noise, outliers, and scale introduced to the dataset, as well as assess the difficulty of the trials.

Author Response

Response to reviewers

We gratefully thank the editor and all reviewers for their time spent making their constructive remarks and useful suggestions, which has significantly raised the quality of the manuscript and enabled us to improve the manuscript. Each suggested revision and comment, brought by the reviewers was accurately incorporated and considered. Specific revisions and responses to each comment are provided in detail below.

Q1. The literature review section of this article is quite deficient. After the introduction, the authors must create a separate section to describe in depth the state of the relevant literature in relation to their research. For instance, the authors only addressed the alternative strategy based on the SVD method, but there are other more modern approaches, such as those based on feature histogram, PCA, triangle similarity ration consistency, as well as normal vector and particle swarm optimization. Consequently, why were these existing methods not discussed?

We gratefully appreciate for your advice. The relevant literature added are as follows(in paragraph 4 of the Introduction Section)

In response to the above problems, scholars have proposed various solutions. The PFH(point feature histogram) algorithm proposed by Radu computes the registration results by extracting an optimal set that best characterizes a given point cloud through feature histograms. This method is insensitive to outliers, but the computational efficiency is generally low due to the high computational dimension(16D). Baowei utilizes the PCA(Principal Component Analysis) algorithm to calculate the covariance matrix of the source and target point clouds, and finds the rotation and transformation matrices between the source and target point clouds based on the covariance matrix. This method works well in the case of scale transformation, but the computation time is also long, taking several minutes to compute 100,000 points. In addition, Xuyan's proposed method based on triangle similarity ratio consistency uses LASH(Local Angle Statistics Histogram) to detect triangle matching points with the same similarity for computing multiple transformations between two point clouds, which has been proved to be robust to noise. Xu's method applys NVP (normal vector and particle swarm optimization) algorithm to find the correspondence points and employs quaternion method to align the correspondence points, which achieves high accuracy even when some data is lost. Yet, this method generally consumes long time when evaluating multiple transformation matrices.

Q2. Due to the lack of relevant literature, it is difficult to comprehend the basis for the ideas presented in this article. Are the histogram and vector operations utilized in this study completely novel? I have my doubts, as there are already studies that apply these methods, so what is genuinely novel about this paper?

Your suggestion is very helpful to improve the quality of this article. The novelty of this paper was added to the 6th paragraph of the Introduction Section

Compared to existing methods, the novelty of this paper is: (i) Other methods address the scale  and outlier problem in terms of feature extraction, and our method addresses both of these problems in the computation of the transformation matrix. (ii) We calculated the scale factor, rotation axis vector, rotation angle and displacement vector separately using vector and histogram methods to reduce the error interactions of these variables. (iii) We make full use of the relationship between vectors (Equation 5 - 8) to calculate the rotation axis vector and the rotation angle. (iv) Even under Gaussian noise, the distribution of Rodrigues parameters will not be the standard positive-terrestrial distribution, which makes the error of methods such as averaging very large, while our method uses histograms to greatly avoid this systematic error and provides a way for others when finding the maximum likelihood variable for non-Gaussian distributions.

Q3. In the results section, please include the images of the two point clouds for each case study. The authors only provided images for Case 1 and partially for Case 2. Please provide images for all case studies associated with each point cloud. By giving these images, readers will be able to comprehend the amount of noise, outliers, and scale introduced to the dataset, as well as assess the difficulty of the trials.

We totally agree with your suggestion. We have added images of the destination point cloud for each case. The destination point cloud without scale transformation, without noise and without outliers in case 1 is shown in Figure. 2. The destination point cloud without scale transformation, with Gaussian noise and without outliers in case 2 is shown in Figure. 4. The destination point cloud with scale transformation, with Gaussian noise and no outliers in case 3 is shown in Figure. 6. The destination point cloud with Gaussian noise and outliers in case 4 is shown in Figure. 9.

Finally, we would like to thank the referee again for taking the time to review our manuscript.

Round 2

Reviewer 1 Report

The paper is modified as per the comments and I am satisfied.